# Energy Consumption of Water Running and Cycling at Four Exercise Intensities

**DOI:** 10.3390/sports10060090

**Published:** 2022-06-08

**Authors:** Sabrina Demarie, Emanuele Chirico, Cecilia Bratta, Cristina Cortis

**Affiliations:** 1Department of Human Movement and Sport Sciences, Università degli Studi di Roma ″Foro Italico″, 00135 Roma, Italy; e.chirico@studenti.uniroma4.it; 2Department of Human Sciences, Society and Health, University of Cassino and Lazio Meridionale, 03043 Cassino, Italy; cecilia.bratta@studenti.unicas.it (C.B.); c.cortis@unicas.it (C.C.)

**Keywords:** energy cost, oxygen consumption, training prescription, fitness, water

## Abstract

Water exercise provides a workload in every direction of motion for training in a reduced impact environment. The selection of an appropriate physical activity and an individual exercise prescription are essential to obtain training effects. The aim of the present study was to determine individualised relative exercise intensities at four speeds of motion for water cycling and water running. Running was tested both in buoyancy and with the feet in contact with the bottom of the pool. To this purpose, gas exchanges, heart rate, and blood lactate were measured in each test session. Fourteen active, healthy females (23.2 ± 1.6 years) underwent a dry land maximal incremental protocol to exhaustion on a treadmill and hydrobike (HB); they engaged in water running with ground contact (RC) and water running suspended (RS) tests in a swimming pool at 30, 40, 50, and 60 cycles per minute (cpm), submerged at the individual xiphoid level. The four motion speeds of the three water exercise modalities ranged from 50% to 95% of the maximal heart rate and the maximal oxygen uptake, representing a moderate-to-vigorous training stimulus. RS elicited the lowest oxygen consumption, whereas HB demanded the significantly highest oxygen consumption and presented the highest blood lactate accumulation, with vigorous intensity being reached at 50 cpm and near maximal intensity at 60 cpm. It appears that water cycling could be more suitable for athletic training, whereas water running could be more appropriate for health and fitness purposes.

## 1. Introduction

Water exercise is defined as an upright exercise in shallow or deep water with the aim of improving fitness and training statuses in a reduced impact environment. Water-based activities have gained popularity and are considered valuable alternatives to traditional physical activities either for people with low levels of physical fitness or who are affected by chronic diseases, but also for athletes [1,2,3]. The specific properties of water, in particular buoyancy, which reduces the effects of weight bearing on skeletal joints, and the great density of water, which provides workloads along all ranges of motion, could make this environment suitable for various types of participants [4]. Previous studies on water exercise programmes have reported several aerobic, metabolic, musculoskeletal, cardiorespiratory, and psychological benefits in participants of water-based programmes [5,6].

Hydrostatic pressure and water temperature are the major modulators of physiological responses to the aquatic environment. Typically, swimming pools have water temperatures ranging from 27 to 31.8 °C. Water temperatures closer to the higher threshold seem to limit the tolerance to exercise, likely due to the increased thermal load. Immersion in waters with temperatures closer to the lower threshold results in the redistribution of blood volume, an increased peripheral vascular resistance, and central hypervolemia, likely activated to respond to the heat loss [7,8]. The selection of an appropriate physical activity in terms of exercise modality, intensity, frequency, and duration is essential to obtain training effects and to ensure exercise adherence [9,10,11]. According to the overload principle of training, exercise below a minimum intensity, or threshold, will not challenge the athlete enough to result in improvements in physiological parameters. Direct measurements of the heart rate and oxygen uptake are recommended for an individualised exercise prescription. However, the minimum threshold of intensity required for benefit is related to the initial state of fitness and conditioning of the subjects. Therefore, for individual exercise prescription, a relative measure of intensity (i.e., the energy cost of the activity relative to the individual’s maximal capacity) is considered the most appropriate [12]. As has been proven for a long time, exercise evaluations for training prescriptions should closely reproduce the subject’s specific sport activity [13]. Thus, it is important to focus on the underlying physics and biomechanics of exercising in the water to better achieve the desired physiological, metabolic, and psychological outcomes and to obtain specific and valid measurements [14,15]. However, water’s properties, such as temperature, buoyancy, hydrostatic pressure, specific gravity, and drag, affect every participant differently based on limb length, body composition, and fat deposition, making it difficult to accurately assess the various components of a water-exercise protocol [2]. Biomechanical and physiological assessments are more difficult in the water than on land; for example, video analyses are greatly hindered by the water refraction and by the water turbulence disturbances [16,17]. Methods that measure perceived effort and the pleasantness of exercise can be used to modulate or refine the prescribed exercise’s intensity, but the accuracy of this approach is insufficient for it to be utilised as a primary method of exercise prescription [12]. The indirect evaluation of physiological thresholds, such as the critical velocity of swimmers and water polo players, has been obtained with simple chronometrical measurements [18,19]. In swimming, velocity and acceleration assessments have been made using inertial sensors to indirectly estimate the anaerobic capacity and to assess kinematic variables [20]. In typical water fitness classes, participants perform complex and highly variable movement sequences, and the intensity of the exercises can vary quite considerably throughout the lesson [21]. The exercise’s intensity can be controlled by setting both the rhythm and the range of execution during aquatic resistance movements [22]. As the speed increases, the higher drag force leads to a rise in muscle activation needed to overcome the water resistance [23]. When walking on a treadmill with subjective speed control, the muscle activation of the lower limbs in the water is higher than on land [3]. When the underwater gait is performed on a treadmill and the speed is controlled, lower muscle activation can be observed compared to the gait on land, regardless of speed [24]. This allows each person to adjust their own effort by altering the range of motion whilst maintaining a constant rhythm. In swimming, water running, and cycling—where the cyclic motion helps to maintain a fixed, predetermined rhythm—it could be easier than in choreographed water fitness lessons to set and maintain a programmed exercise’s intensity. However, the range of motion cannot be modified when pedalling on a bike in the water, but it can be easily adjusted when water running, or when performing buoyancy exercises. This makes each of these exercise modalities more or less viable depending on the targeted population and the desired training outcome. During speed-controlled water exercises, the direct assessment of heart rate and oxygen uptake levels relative to the individual’s maximal values could provide useful indications for individualised exercise prescriptions.

The aim of the present study was to assess the maximal aerobic power on land and to determine individualised relative exercise intensities for water cycling and water running, both in buoyancy and in contact with the bottom of the pool, at the same water depth, in young and healthy active females.

## 2. Materials and Methods

### 2.1. Subjects

Fourteen female college students (Mean ± SD age 23.2 ± 1.6 years, weight 58.2 ± 5.0 kg, stature 161.4 ± 6.6 cm, BMI 22.4 ± 1.8) who attend water fitness courses participated in the study. All participants received written and oral information about the purpose of the study, of their rights as study participants, and of the anonymity of their data and provided written informed consent. The project received approval from the institutional review board (CAR-IRB 122/2022).

### 2.2. Procedure

All fourteen female college students underwent four experimental sessions at the same time of the day (around 18:00 PM). Since subjects participated in generic water fitness classes, they were not supposed to have undergone any specific training adaptation; therefore, to ensure that maximal values were attained, the maximal aerobic power was assessed with a standardised incremental treadmill procedure [25]. The maximal incremental treadmill (Skillrun, Technogym, Cesena, Italy) tests were performed during the first experimental session in a gym close to the swimming pool. Hydrobike (HB), running with ground contact (RC), and suspended running (RS) tests were carried out on separate days in a random order. All water exercise tests took place in the same 25 m swimming pool with a constant 27 °C water temperature; each subject was positioned at a pool depth corresponding to the individual xiphoid level. The xiphoid process depth of body immersion was chosen because it offers more stability compared to the immersion at the clavicle depth [26]. HB was performed on an Aquabike Piscina (AquaNess, Sommerau, France), RC consisted of running with the feet in contact with the bottom of the pool, and RS was conducted using a buoyancy belt (Aqua Fitness Jogbelt, Speedo, Nottingham, UK. During HB, the hands gripped the handlebars, whereas during the two running tests the upper limbs had no constraints. All the tests were preceded by a 15 min light- intensity warmup outside of the water, on a cycle ergometer, at 20 cycles per min (cpm) [21]. After the warmup, the subjects were equipped for breath-by-breath gas analysis (K4 b2, Cosmed, Rome, Italy). Gas analysers were calibrated immediately before each test with a known concentration of oxygen and carbon dioxide and the flow meter was calibrated with a 3-l syringe. The heart rate was continuously monitored (Sport Tester, Polar Electro, Helsinki, Finland). For the determination of peak blood lactate accumulation, blood samples were collected from the earlobe at rest, at the third, sixth, and ninth minute of passive recovery, whilst they sat at rest. The measurement of the blood lactate value was carried out immediately after the collection.

### 2.3. Maximal Test

The starting velocity of the maximal incremental treadmill test was 7 km/h and it was increased by 1 km/h every minute until the subject voluntarily stopped due to exhaustion. The subjects were verbally encouraged to give their maximal effort, starting from the moment the respiratory exchange ratio reached a value higher than 0.9. The achievement and calculations of maximal values were ensured following a procedure described previously [27]. Gas exchanges were measured at the mouth by a portable breath-by-breath metabolimeter (K4b2, Cosmed, Roma, Italy) and the heart rate was continuously recorded (Polar, Electro, Helsinki, Finland). Later, collected data were averaged every 5 s. Blood samples were taken from the earlobe after a 15 min warmup and 3, 6, and 9 min after the end of the test; the lactate concentration assessment was carried out immediately after collection by a portable lactacidometer (LactatePro2, Arkray, Kyoto, Japan).

### 2.4. Water Tests

All the subjects were able to swim and were familiarised with hydrobikes, as well as with shallow- and deep-water running. An experienced fitness coach carefully instructed the subjects on the appropriate cycling and water-running techniques and approved the technique of all the subjects, as previously described [28]. Subjects were allowed to move forwards as they ran, both in deep and shallow water. The water tests consisted of cycling or running at 30, 40, 50, and 60 cpm, indicated by an acoustic pre-registered signal. Each stride or cycling frequency was performed for 5 min with a 1 min rest. During the 5 min exercise, the oxygen uptake was assessed breath-by-breath by a portable metabolimeter through a mouthpiece designed for use during swimming that was attached to a lightweight helmet (K4b2 and Aquatrainer, Cosmed, Italy); the gas analyser contained in a water-resistant bag was loaded on a rack located immediately over the head of the participant and carried by a person so that it would closely follow the subject. At the end of the swimming pool, the subject made a pre-instructed outwards turn. The heart rate was continuously recorded (Polar, Electro, Finland); after the collection, data were averaged every 5 s. Before starting the tests and during the 1 min rest periods amongst repetitions, blood samples were collected from the earlobe for blood lactate concentration measurements.

### 2.5. Data Analysis

Oxygen uptake (VO_2_) was recorded for each breath and then averaged every 5 s. The oxygen consumption in mL·kg^−1^·min^−1^ was calculated as the VO_2_ mean value of the last minute of each step. The net accumulation of blood lactate (bLA) was obtained by subtracting the rest from the peak values attained during the recovery phase of each test. Heart rate (HR) was calculated as the mean value of the last minute of each step. All the values have been reported as absolute values and as percentages of the maximal values measured during the dry land incremental maximal test (VO_2_% VO_2_max and HR % HRmax).

### 2.6. Statistical Analysis

For all data, descriptive statistics were presented, and distributions were verified to identify potential outliers. The distribution was checked and whether assumptions of normality had been violated was verified with a Shapiro–Wilk test. In the case of a skewed distribution, differences amongst exercise modalities and intensities were analysed through a Kruskal–Wallis test as a non-parametric method. In the case of a normal distribution, the differences were tested through a one-way Analysis of Variance (ANOVA) for repeated measures and by the Bonferroni post hoc. All statistical tests were performed using the software SPSS v20.0 (SPSS, Chicago, IL, USA) and Microsoft Excel 2010. The level of significance was set at *p* < 0.05.

## 3. Results

All data were normally distributed. During the treadmill tests, maximal measured values were VO_2_max 2621.3 ± 240.7 (mL·min^−1^), VO_2_max 45.4 ± 5.4 (mL·kg^−1^·min^−1^), HR max 196.8 ± 10.6 (bpm) and La peak 10.4 ± 0.9 (mM/L).

Table 1 shows that the heart rate and oxygen consumption of suspended running resulted in significantly lower rates than the hydrobike’s in each exercise intensity. Suspended running also presented a significantly lower heart rate than running, with ground contact at 50 and 60 cpm, and a significantly lower oxygen consumption at 40, 50 and 60 cpm. Blood lactate accumulations, calculated by subtracting the rest from the peak values obtained during the recovery phase, resulted in significantly lower rates for suspended running as compared to the hydrobike at 50 and 60 cpm only. In the HB exercise, bLa exceeded 4 mM/L, even at 50 cpm, whereas water running presented bLa above 4 mM/L only at 60 cpm.

Figure 1 depicts the heart rate, oxygen consumption and lactate values in percentages of maximal values measured during the dry-land maximal cycle ergometer test. At the higher exercise intensities (50 and 60 cpm), all three parameters were statistically higher for the hydrobike exercise compared to both water-running modalities. Based on the American College of Sports Medicine guidelines for prescribing exercise [12], percentages of maximal heart rates of all exercise modalities elicited moderate intensities at 40 cpm, and vigorous at 60 cpm. At 50 cpm, both water-running modes were still in the moderate intensity domain, whereas the hydrobike presented a vigorous intensity. Based on oxygen uptake, in hydrobike exercises, vigorous intensity was already reached at 40 cpm in all exercise modalities, and near maximal intensities were recorded at 60 cpm.

## 4. Discussion

The aim of this study was to measure the heart-rate response and the oxygen consumption of water-based cycling and two running modalities, suspended in buoyancy and with the feet in contact with the bottom of the pool, at four different motion frequencies, whilst at the xiphoid level of immersion.

Heart rate, oxygen uptake, and blood lactate accumulation responses measured throughout the 5 min trial of the four motion speeds in the three exercise modalities ranged from 50% to 95% of the maximal values and are included within the moderate-to- vigorous and near-maximal training stimuli to improve cardiorespiratory fitness, as defined by the American College of Sports Medicine [12]. To differentiate amongst the effects of the three modalities of exercise, oxygen consumption as a percentage of maximal aerobic power provided the assessment of the relative exercise intensity for each motion frequency. Running suspended required the least cardiometabolic effort; blood lactate accumulation was above resting values, ranging from 52% to 76% of HRmax, from 49% to 72% of VO_2_max, with a blood lactate accumulation from 2 to 5 mM/L. Its heart rate values were included in the moderate intensity domain at all exercise frequencies, whilst VO_2_% reached the vigorous intensity domain at 60 cpm. Insignificantly higher effort was demanded by running whilst in contact with the bottom of the pool, but heart rate at 60 cpm, and VO_2_ at both 50 cpm and 60 cpm brought the intensity into the vigorous domain, with a blood lactate accumulation of roughly 6 mM/L. Cycling at the same water level and rhythm of the two running modes demanded the significantly highest oxygen consumption, and had the highest blood lactate accumulation, with vigorous intensity being reached at 50 cpm and near maximal intensity at 60 cpm for both the heart rate and VO_2_ parameters, with a blood lactate accumulation over 4.5 mM/L at 50 cpm and around 7 mM/L at 60 cpm. These results agree with those reported throughout a 40 min water exercise trial, when the greatest motion speed possible was reached whilst maintaining a full range of motion, and the heart rate reached 90% of maximal HR. The high cardiorespiratory response was most likely due to the optimization of water resistance at maximal speeds and full ranges of motion during the downwards action against buoyancy, and upwards work pushing against the bottom of the pool [2]. Running suspended and in contact with the bottom of the pool allowed more free-motion amplitude than cycling; consequently, at the same speed of motion, they elicited significantly different intensities of exercise. Consistently, when the heart rate and blood lactate responses in young and healthy women performing the same routine of aerobic exercise on land and in the water were compared, the variability in physiological measures amongst participants has been reported to decrease considerably when moving from land to the water environment [7]. In water running, the possibility for all the participants to regulate their individual metabolic requirement for each rhythm of exercise could lead to a greater exercise adherence for fitness and health purposes [4]. This might be of great importance for the aim of fitness activities, where people often have different ages and levels of ability, implying that water exercise could be an effective practice for a controlled physiological response in heterogeneous groups. It could also be an important point to consider when following guidelines for exercise prescription, such as those of the American College of Sports Medicine, to improve cardiorespiratory fitness or to reduce the risk of chronic degenerative diseases [12,29].

On the other hand, when running in the water at the xiphoid level of immersion, the resistance of the water could hinder the attainment of ample ranges of motion and vigorous intensities of exercise. Therefore, it has been suggested that exercise at a body immersion deeper than the xiphoid level does not provide a sufficiently stable surface for a significant force to be applied, whereas when the body is less immersed, it suffers less imbalance, so a greater force can be applied [26]. In accordance with this, it has been reported that the deeper the water, the less the physiological demand of the exercise [7]. It can thus be suggested that, without contact with a fixed-point, a high intensity of exercise in the water can be less easily reached. Indeed, the quantification and regulation of the work intensity during water-based activities is still problematic. In water-based activities performed without specific equipment or aquatic devices, the actual mechanical work cannot be directly measured, since the work needed to overcome the water resistance encountered is dependent on the limb’s mass moving through it; furthermore, the range of motion is difficult to control and measure [4,21]. The assessment of the motion’s amplitude should be addressed, utilising video recordings that are commonly employed for the calculations of stroke rate, stroke length, and the assessment of the general characteristics of a swimmer’s style. However, although the video analysis approach is successfully employed as an analytical tool in many sport disciplines, its application for the assessment of submerged water exercise is limited by the turbulence caused by the water, which affects the view of the anatomical points of interest. Inertial sensors have already been proven useful for direct kinematic and indirect physiological evaluations of swimmers; multiple sensor-based measurements of swimmers’ acceleration profiles have the potential to offer significant advances in coaching techniques over the traditional video-based approach [20]. Recording of tri-axial accelerations and the velocities of ankles and wrists could be a practical method to determine the speed of the limbs’ motions and to estimate the intensity of various exercises in the water, even complex choreographed ones. Of practical significance, instructors should recognise the importance of proper technique, range, and speed of motion during water exercise, which may influence an individual’s ability to achieve the desired intensity [2].

Aquatic cross training has been suggested to be a useful method for athletes, as it offers a unique training stimulus that can maintain aerobic performance whilst decreasing the stress of the training environment [1,28]. As suggested by the present results, at the same, highest speed of exercise (60 cpm), the suspended running exercise with a completely free range of motion stood in the moderate intensity domain, the running in contact with the bottom of the pool exercise with a partially limited range of motion passed into the vigorous domain, and the cycling exercise with a fixed range of motion brought the intensity close to the maximal domain. Therefore, water running would require very high speeds of motion to reach vigorous and near maximal intensities, whereas cycling in the water permits them to be attained already at 50 cpm and 60 cpm. In accordance with this, it has been proven that pedalling in the water can allow very high exercise intensities to be reached and that excessive pedalling speeds can be avoided by adding a resistance tool to the pedals [29]. Due to the possibility of controlling a motion’s rhythm and amplitude, water cycling appears to be a more suitable exercise modality over water running for athletes’ training purposes. It allows them to respect the predetermined load and to practise at vigorous exercise intensities. For athletes and coaches, the determination of exercise intensity plays a vital role for training planning purposes [9]. Methods for prescribing exercise intensity include percentages of maximal and submaximal reference measurements derived from graded exercise tests that are indicative of different physiological thresholds [30]. Incremental maximal test procedures are seldom used in water sports due to the well-known complications posed by the aquatic environment [19,31,32]. As an alternative, critical power and critical velocity, which are empirical parameters that define the relationship between power output or speed and time-to-exhaustion, could be recommended as versed protocols for water sports’ testing [30,33]. It has been shown that critical velocity is positively correlated with the swimming velocity at the maximum lactate steady state, calculated with graded incremental tests. Furthermore, it enables the desired training intensity to be chosen [18,34]. Pedalling in the water could be performed at increasing workloads to the point of exhaustion, so that the hyperbolic relationship between power output and time to exhaustion could reveal the highest sustainable work rate that enables lactate to remain in steady-state conditions. With this method, the heavy and severe domains of exercise can be precisely assessed for training prescription purposes [35]. It could therefore be suggested that water cycling could represent an appropriate exercise mode for training purposes, as it enables athletes to reach vigorous intensities of exercise.

Amongst the limitations of the present study are the restricted number and characterisation of the participants, being representative of Caucasian, young and healthy females only. Moreover, exercises have been tested at a single water depth, and at submaximal speeds of motion only. Deeper knowledge on the topic of water exercise could be obtained by researching different groups of subjects, water depths, and speeds of exercise.

In conclusion, the three water exercise modalities analysed, i.e., cycling, suspended running, and running in contact with the bottom of the pool, provide different intensities at the same speed of motion. Water cycling elicited the highest energy consumption, whereas suspended water running required the least cardiometabolic effort and presented the lowest blood lactate accumulation.

## Figures and Tables

**Figure 1 sports-10-00090-f001:**
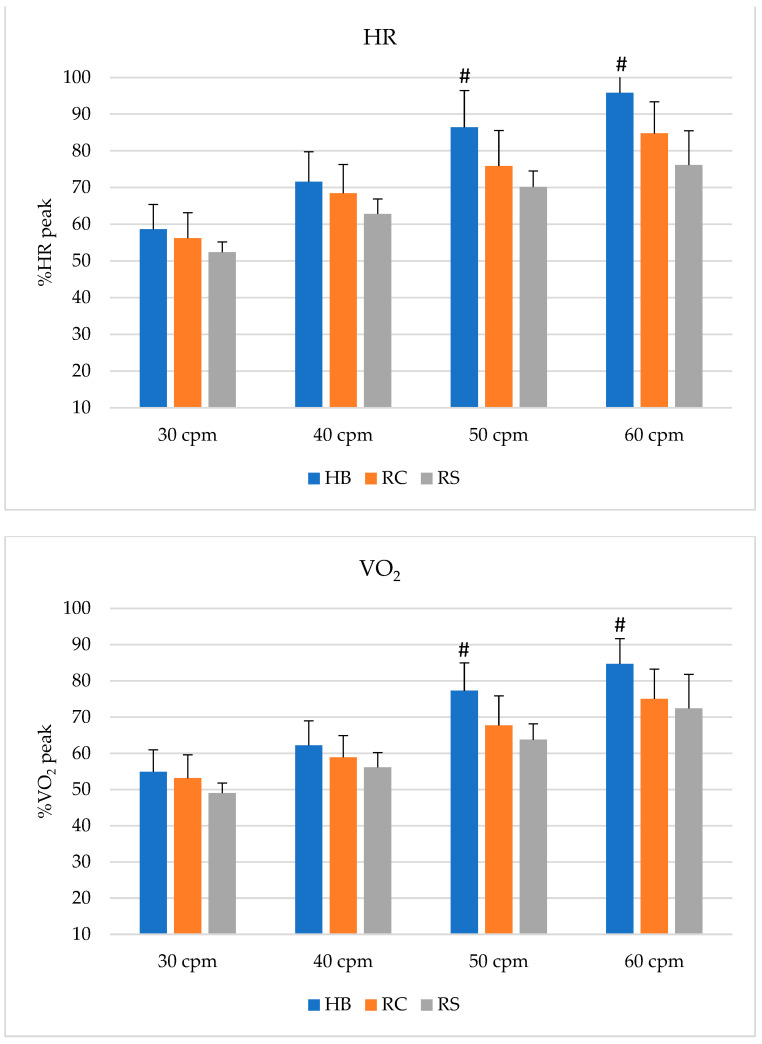
Heart rate (HR) and oxygen consumption (VO_2_) in percentages of maximal values; blood lactate accumulation above resting levels (bLa) in absolute values and in percentages of maximal values for hydrobike (HB), running in contact with the bottom of the pool (RC), and suspended running (RS) tests. # = *p* < 0.05 between HB and both RS and RC.

**Table 1 sports-10-00090-t001:** Mean and standard deviation values for each exercise intensity and modality.

		30 cpm	40 cpm	50 cpm	60 cpm
HR (bpm)	HB	115 ± 12	140 ± 15	170 ± 19 ***	188 ± 18 ***
	RC	110 ± 13	134 ± 15	149 ± 18 **	166 ± 17 **
	RS	103 ± 6 *	123 ± 11 *	138 ± 8 *	149 ± 188 *
VO_2_ (mL·kg^−1^·min^−1^)	HB	24.8 ± 3.2	28.1 ± 3.5	34.8 ± 3.4 ***	38.1 ± 3.6 ***
	RC	23.8 ± 2.2	26.5 ± 2.5 **	30.3 ± 2.6 **	33.7 ± 2.9 **
	RS	22.0 ± 2.0 *	25.3 ± 3.3 *	28.7 ± 3.3 *	32.6 ± 3.5 *
bLa (mM/L)	HB	2.2 ± 0.4	2.8 ± 0.2	4.6 ± 0.9 ***	6.9 ± 0.7 ***
	RC	2.2 ± 0.3	2.8 ± 0.3	3.6 ± 0.5	6.1 ± 0.6
	RS	2.1 ± 0.2	2.6 ± 0.1	3.2 ± 0.4 *	5.0 ± 0.9 *

* = *p* < 0.05 between RS and HB; ** = *p* < 0.05 between RS and RC; *** = *p* < 0.05 between HB and RC. HR: heart rate; VO_2_: oxygen consumption; bLa: blood lactate accumulation. RS: running suspended; RC: running in contact with the bottom of the pool; HB: hydrobike.

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
