# Peer review of "Energy Consumption of Water Running and Cycling at Four Exercise Intensities"

_sports, 2022, doi:10.3390/sports10060090_

Round 1

Reviewer 1 Report

External load is used in the title but not clarified in the paper. Please provide clarification in the manuscript as external load can be many things.

Please provide the water temperature.

Please provide more detail on the time of day for testing. Was that for each individual the same?

Please provide the recovery observations, e.g. see Ls 119-120.

L10. Please clarify/revise the first sentence of the abstract. The phrase “all range of movements for training “ is vague.

L12. Is there evidence in the literature on injuries with water exercise? See also L41.

L13. Movement is changing location. Is stationary cycling described as a movement. Please revise.

L15. I suggest to provide additional subject characteristics, e.g. BMI.

L20. Please clarify 50-95% of which maximal values.

L33. Change “chronical” to “chronic”.

L38. Please clarify whether this is blood flow to the muscles. In addition, do all water temperatures enhance blood flow? Can you be more specific here as I assume water exercise responses have been examined with different water temperatures.

L49. Please change “[12]. ].”.

L73. Please change “muscle activation results higher than on land” to “muscle activation is higher than on land”

L94. Provided units for age.

L113. In the RS condition, it is stated “with free movement of the upper limbs”. Was there not free movement of the lower limbs as it the RS is described as running. Please clarify.

L114. What was the rationale of a rpm of 20 which seems to be very slow.

L117. Why was the K4 calibrated after the test?

Ls 119-120. Why were blood samples taken during the recovery. Please justify and provide the values.

L125. How did you know when the last minute was happening as only verbal encouragement was provided in the last minute.

L151. Energy cost by oxygen consumption is mostly valid when in steady-state. Did the participants reach steady state with respect to heart rate and oxygen in those 5 min bouts. The authors may want to consider energy cost is appropriate description or just referring to the oxygen consumption of the exercises.

L167. Please change “All data resulted” to “All data were”.

L168. Please provide the parameter before the values, e.g. VO2max: 2621.3 ± 240.7 ml/min. Do this throughout the manuscript.

L168. Please express heart rate without decimal places.

L168. Please change “ml/kg/min” to “ml·kg-1·min-1“ or “ml/(kg·min)” throughout the manuscript.

Table 1. VO2 seems to be relative values so change “l” to “ml”.

Table 1. I suggest to clarify in the results section that the lactate here is as described in the methods “The net accumulation of blood lactate (bLA) was obtained subtracting the resting value from that attained during each recovery phase of the water exercise”. Why not just providing all absolute lactate values including those at rest.

In Figure 1, are the observations for lactate the accumulated values with respective to the accumulated maximal value. I suggest to analyse the relative values for lactate with absolute observations of lactate.

L193. Please provide a description of the abbreviations in the figure legend.

In the figure 1 legend, please clarify what is meant by the significance sign.

In Figure 1, please provide mean and SD values.

In Figure 1, please reconsider whether the y-axis scale for heart rate, oxygen consumption and lactate needs to start at 0. The value of zero can be obtained but there is not much life!

The quality of the figures must be improved.

L208. Metabolic responses are not provided. Please revise. Do you mean the oxygen consumption?

Ls 212-215. Is this accumulation of lactate above resting values? Please clarify.

L306. Why was the sample size a limitation. Please clarify whether the study was underpowered.

The format of references is not consistent. Please consult the author guidelines of the journal.

Author Response

Dear Reviewer,

thank you for your comments and suggestions, hoping to have positively fulfilled all your request, we have listed our answers point by point.

External load is used in the title but not clarified in the paper. Please provide clarification in the manuscript as external load can be many things.

ANSWER – Thank you for pointing out this inconsistency, we modified the title of the manuscript in “Energy consumption of water running and cycling at four exercise intensities”, and we used the term “motion” instead of “movement” in the text.

Please provide the water temperature.

ANSWER – line 106: All water exercise tests took place in the same 25 m swimming pool with a constant 27°C water temperature.

Please provide more detail on the time of day for testing. Was that for each individual the same?

ANSWER – lines 98-99: All fourteen female college students underwent four experimental sessions at the same time of the day (around 18:00 PM).

Please provide the recovery observations, e.g. see Ls 119-120.

ANSWER – line 119: Blood lactate determination from the earlobe at rest, and at the third, sixth, and ninth minute of the passive recovery while they sat at rest, was carried out immediately after the collection.

L10. Please clarify/revise the first sentence of the abstract. The phrase “all range of movements for training “ is vague.

ANSWER – line 10: Water exercise provides workload in every direction of movements for training in a reduced impact environment.

L12. Is there evidence in the literature on injuries with water exercise? See also L41.

ANSWER – overuse damages are frequent in swimmers, water polo player and triathletes, but not in water exercises, therefore we eliminated the term injury from the text. line 12: The selection of an appropriate physical activity in terms of exercise modality, intensity, frequency, and duration is essential to obtain training effects and to ensure exercise adherence [9-11].

L13. Movement is changing location. Is stationary cycling described as a movement. Please revise.

ANSWER – thank for this suggestion, since cycling is involved in all our data analysis we substituted the term “movement” with “motion” in the title and along the text.

L15. I suggest to provide additional subject characteristics, e.g. BMI.

ANSWER – we added BMI 22.4 ± 1.8 in the subject’s description.

L20. Please clarify 50-95% of which maximal values.

ANSWER – we added the information: ranged from 50% to 95% of maximal heart rate and maximal oxygen uptake.

L33. Change “chronical” to “chronic”.

ANSWER – the change has been made.

L38. Please clarify whether this is blood flow to the muscles. In addition, do all water temperatures enhance blood flow? Can you be more specific here as I assume water exercise responses have been examined with different water temperatures.

ANSWER – we added a brief explanation on blood flow redistribution in humans due to immersion in the water: Hydrostatic pressure and water temperature are the major modulators of physiological responses to the aquatic environment. Typically, swimming pools have water temperatures ranging from 27 to 31.8°C. Water temperatures closer to the higher threshold seem to limit the tolerance to exercise, likely due to the increased thermal load. Immersion in waters with temperatures closer to the lower threshold results in the redistribution of blood volume, an increased peripheral vascular resistance and central hypervolemia, likely activated to respond to the heat loss [7,8].

L49. Please change “[12]. ].”.

ANSWER – the change has been made.

L73. Please change “muscle activation results higher than on land” to “muscle activation is higher than on land”

ANSWER – the change has been made.

L94. Provided units for age.

ANSWER – years has been added.

L113. In the RS condition, it is stated “with free movement of the upper limbs”. Was there not free movement of the lower limbs as it the RS is described as running. Please clarify.

ANSWER – test description has been modified: HB was performed on an Aquabike Piscina (AquaNess, France), RC consisted of running with the feet in contact with the bottom of the pool, RS was conducted using a buoyancy belt (Aqua Fitness Jogbelt, Speedo, USA). During HB the hands gripped the handlebars, whilst during the two running tests the upper limbs had no constraints.

L114. What was the rationale of a rpm of 20 which seems to be very slow.

ANSWER –  as indicated by Raffaelli 2010 (https://doi.org/10.1007/s00421-010-1419-5) the warm up phase was carried out asking the subjects to perform exercises at frequencies that correspond, for physically active women (26.4 ± 3.8 years) to a ‘‘light’’ intensity level (20–39 % of VO2max).

L117. Why was the K4 calibrated after the test?

ANSWER – K4 was calibrated after the tests because we carried out consecutive measures (“after” test 1 corresponded to “before” test 2), but, as you pointed out, this description is redundant and confusing, so the calibration is now indicated only before each test.

Ls 119-120. Why were blood samples taken during the recovery. Please justify and provide the values.

ANSWER – blood samples were collected during the recovery phase of each test to detect the peak value that we reported in Tables and Figure. We modified the description in the text as follows: For the determination of peak blood lactate accumulation, blood samples were collected from the earlobe at rest, and at the third, sixth, and ninth minute of the passive recovery while they sat at rest. The measurement of the blood lactate value was carried out immediately after the collection.

L125. How did you know when the last minute was happening as only verbal encouragement was provided in the last minute.

ANSWER – a more precise indication of the starting of verbal encouraging has been provided: The subjects were verbally encouraged to give their maximal effort, starting from the moment the respiratory exchange ratio reached a value higher than 0.9.

L151. Energy cost by oxygen consumption is mostly valid when in steady state. Did the participants reach steady state with respect to heart rate and oxygen in those 5 min bouts. The authors may want to consider energy cost is appropriate description or just referring to the oxygen consumption of the exercises.

ANSWER – following your suggestion we changed the term “cost” with “consumption”.

L167. Please change “All data resulted” to “All data were”.

ANSWER – the change has been made.

L168. Please provide the parameter before the values, e.g. VO2max: 2621.3 ± 240.7 ml/min. Do this throughout the manuscript.

ANSWER – the changes have been made.

L168. Please express heart rate without decimal places.

ANSWER – the changes have been made.

L168. Please change “ml/kg/min” to “ml·kg-1·min-1“ or “ml/(kg·min)” throughout the manuscript.

ANSWER – ml/kg/min” has been changed to “ml·kg-1·min-1”.

Table 1. VO2 seems to be relative values so change “l” to “ml”.

ANSWER – the change has been made.

Table 1. I suggest to clarify in the results section that the lactate here is as described in the methods “Blood lactate accumulations, calculated subtracting the rest from the peak values obtained during the recovery phase”. Why not just providing all absolute lactate values including those at rest.

ANSWER – the explanation on how blood lactate accumulation was calculated has been added to the description of Table 1. We did not report all gross lactate values because displaying 4 values instead of 1 could have been redundant and confusing. Moreover, peak of blood lactate during recovery does not appear at the same time for all subject and for all exercise intensities.

In Figure 1, are the observations for lactate the accumulated values with respective to the accumulated maximal value. I suggest to analyse the relative values for lactate with absolute observations of lactate.

ANSWER – in figure 1 the depiction of lactate values in absolute has been added.

L193. Please provide a description of the abbreviations in the figure legend.

ANSWER – the description has been provided as follows: Heart rate (HR) and oxygen consumption (VO2) in percentages of maximal values; blood lactate accumulation above resting levels (bLa) in absolute values and in percentages of maximal values for hydrobike (HB), running in contact with the bottom of the pool (RC) and suspended running (RS) tests.

ANSWER – the meaning has been expressed as follows: In the figure 1 legend, please clarify what is meant by the significance sign.

# =p<0.05 between HB and both RS and RC

In Figure 1, please provide mean and SD values.

ANSWER – the standard deviation values have been added to the figure.

In Figure 1, please reconsider whether the y-axis scale for heart rate, oxygen consumption and lactate needs to start at 0. The value of zero can be obtained but there is not much life!

ANSWER – the minimum values of y-axis scales have been moved to 10%.

The quality of the figures must be improved.

ANSWER – Figure 1 has been modified as follows:

Figure 1. heart rate and oxygen consumption in percentages of maximal values; blood lactate in absolute and in percentages values.

L208. Metabolic responses are not provided. Please revise. Do you mean the oxygen consumption?

ANSWER – “metabolic responses” has been changed with “oxygen consumption”.

Ls 212-215. Is this accumulation of lactate above resting values? Please clarify.

ANSWER – it has been clarified in the text that accumulation of lactate is above resting values.

L306. Why was the sample size a limitation. Please clarify whether the study was underpowered.

ANSWER – what we intended was that our data cannot be inferred to different populations. We tried to clarify it in the text as follows: Amongst the limitations of the present study are the restricted number and characterization of the participants, being representative of Caucasian, young and healthy females only.

The format of references is not consistent. Please consult the author guidelines of the journal.

ANSWER – the format of the references has been adapted to the journal guidelines.

Reviewer 2 Report

In the part of procedure,please clarify whether all 14 female college students participated in the three exercises of HB, RC and RS in this experiment, or were the 14 participants divided into three groups and each group only participated in one of the exercises?

At line 110, the immersion level is described in the article cited here as "Performing the exercise at shallower depths allows greater muscle activation of the extremities. Therefore, if maximum muscle activation of the extremities is required, xiphoid depth is a better choice than clavicle depth". However, I cannot find direct evidence that immersion at the xiphoid level is more stable.

Based on the description from lines 183 – 185, at the exercise intensities of 50cpm and 60 cpm, all three parameters of hydrobike exercise are statistically higher than those of two water running modalities. However, in Table 1,it does not define a symbol to indicate whether there is a significant difference between HB and RC.

Figure 1 may have two areas for improvement. First, the “*” in the figure represents the significant difference between which two groups of data? Does the “*” sign mean the same as the “*” sign in Table 1? Please give a detailed description to facilitate the reader's understanding. In addition, the percentage numbers of maximal values and the ranges of those data in Figure 1 are not marked. It is recommended to redesign the depiction of Figure 1.

This experiment only selected the water depth of immersion at the level of the xiphoid. Is it necessary to add different water depth tests to select the most suitable depth for these three water exercises?

Author Response

Table 1. mean and standard deviation of absolute values for each exercise intensity and modality.

30 cpm

40 cpm

50 cpm

60 cpm

HR (bpm)

HB

115.3 ± 12.8

140.8 ± 15.8

170.0 ± 19.6***

188.6 ± 18.7***

RC

110.5 ± 13.2

134.6 ± 15.1

149.2 ± 18.7**

166.8 ± 17.0**

RS

103.1 ± 6.7*

123.5 ± 11.9*

138.0 ± 8.9*

149.8 ± 188.6*

VO2 (l/kg/min)

HB

24.8 ± 3.2

28.1 ± 3.5

34.8 ± 3.4***

38.1 ± 3.6***

RC

23.8 ± 2.2

26.5 ± 2.5**

30.3 ± 2.6**

33.7 ± 2.9**

RS

22.0 ± 2.0*

25.3 ± 3.3*

28.7 ± 3.3*

32.6 ± 3.5*

bLa (mM/l)

HB

2.2 ± 0.4

2.8 ± 0.2

4.6 ± 0.9***

6.9 ± 0.7***

RC

2.2 ± 0.3

2.8 ± 0.3

3.6 ± 0.5

6.1 ± 0.6

RS

2.1 ± 0.2

2.6 ± 0.1

3.2 ± 0.4*

5.0 ± 0.9*

*=p<0.05 between RS and HB; **=p<0.05 between RS and RC; ***=p<0.05 between HB and RC. HR: heart rate, VO2 energy cost and blood lactate values. RS: running suspended; RC: running in contact with the bottom of the pool; HB: hydrobike.

Figure 1. heart rate and oxygen consumption in percentages of maximal values; blood lactate in absolute and in percentages values.

Heart rate (HR) and oxygen consumption (VO2) in percentages of maximal values; blood lactate accumulation above resting levels (bLa) in absolute values and in percentages of maximal values for hydrobike (HB), running in contact with the bottom of the pool (RC) and suspended running (RS) tests. # =p<0.05 between HB and both RS and RC

Reviewer 3 Report

The topic is interesting and relevant in this area of research. However, the readability of this work needs to be improved. Therefore, involving an English language specialist in this revision is suggested.

Some sentences appear too long and confusing. Try to rephrase and shorten the sentences wherever possible.

The conclusion needs to be written more concisely to make the point. Presenting the main findings of your research should be explained more clearly.

Ln13,14 – “..of movement for water cycling and for water running in buoyancy and in contact with the bottom of the pool.” – Rephrase this part, please.

Ln30 – “or annulled impact environment” - Please check if the terminology is correct and rephrase.

Ln33 – “or for” - This seems unclear. Please correct it.

Ln35 – “could be the basis of this success.” - Please check the meaning of this part of the sentence, and if needed, rephrase accordingly.

Ln49- “.].” – Delete this.

Ln51 – “it is thus” - Please rephrase this as "Thus, it is important"

Ln52 – “produce” - Please consider using another term here.

Ln66-68 - Please consider rephrasing the entire sentence in order to improve readability.

Ln73 – “results” - Please either rephrase the sentence or use another term here.

Ln84-87 - this sentence needs to be rephrased.

Ln120 – “minutes” – minute.

Ln148 – “at the” – from the.

Ln149 – “(LactatePro2….) “- This is already mentioned earlier. Please remove it.

Ln151 – “every” – with each.

Ln158 – statistical analysis - There is no information of statistical software used for this study. Please add.

Ln165 – p-value needs to be lower case letter. p<0.05.

Ln174 – “statistically” – significantly.

Ln175 – “already” - please check if this term is appropriate.

Ln177 – Mean - capital letter.

Ln185 – “with respect to” - compared to.

Ln186 – “guidance” – guidelines.

Ln188 – “still were” – were still.

Ln202 – “xyphoid” – xiphoid.

Ln206 – American College of Sports Medicine – check everywhere in the text and correct.

Ln213-215 - This sentence needs to be rephrased.

Ln223 – “agree” - Please check if this term is appropriate.

Ln224 – “speeds” – speed.

Ln225 – “was” - use another term here, please.

Ln229 - “movement’s” – movement.

Ln234-237 – Rephrasing the sentence is suggested.

Ln244 – “running suspended” – Do you mean suspended running?

Ln245,246 - Rephrasing this part of the sentence is necessary. 

Ln273 – “means”  - Please use "tool" or another similar term.

Ln286 – “predetermined load…” - Please recheck this part of the sentence, and rephrase it to improve readability.

Ln287 – “coached” – coaches.

Ln291 – “tests” – test.

Ln292 – “troubles” - Please use a different term here.

Ln296 – “significantly” – This is not necessary, can be removed.

Ln298 – “bout” – This is not necessary, can be removed.

Ln301 – “in” – Do you mean at steady state?

Ln302-305 – Please rephrase in order to improve readability.

Ln306,307 – “Limits of the present study are, certainly the low number of participants and the whole female representative” - Rephrasing this part is suggested.

Ln309 – “brought about” – Please use another more appropriate term here.

Ln311,312 - Please try to write two sentences here instead of one, and rephrase in order to improve the readability.

Ln316 – “highly fit” - Check if this term is appropriate and rephrase if needed.

Author Response

Dear Reviewer,

thank you for your comments and suggestions, hoping to have positively fulfilled all your request, we have listed our answers point by point.

The topic is interesting and relevant in this area of research. However, the readability of this work needs to be improved. Therefore, involving an English language specialist in this revision is suggested.

Some sentences appear too long and confusing. Try to rephrase and shorten the sentences wherever possible.

The conclusion needs to be written more concisely to make the point. Presenting the main findings of your research should be explained more clearly.

ANSWER – the conclusions have been re written: In conclusion, the three water exercise modalities analysed, i.e., cycling, suspended running and running in contact with the bottom of the pool, provide different intensities at the same speed of motion. Water cycling elicited the highest energy consumption, whilst suspended water running requested the least cardiometabolic effort and presented the lowest blood lactate accumulation.

Ln13,14 – “..of movement for water cycling and for water running in buoyancy and in contact with the bottom of the pool.” – Rephrase this part, please.

ANSWER – the sentence has been rephrased: Aim of the present study was to determine individualized relative exercise intensities at four speeds of movement for water cycling and for water running. Running was tested both in buoyancy and with the feet in contact with the bottom of the pool.

Ln30 – “or annulled impact environment” - Please check if the terminology is correct and rephrase.

ANSWER – the sentence has been rephrased: Water exercise is defined as upright exercise in shallow or deep water with the aim of improving fitness and training status in a reduced impact environment.

Ln33 – “or for” - This seems unclear. Please correct it.

ANSWER – the sentence has been rephrased: Water-based activities have gained popularity and are considered as valuable alternatives to traditional physical activities either for people with low levels of physical fitness or chronical disease, but also for athletes.

Ln35 – “could be the basis of this success.” - Please check the meaning of this part of the sentence, and if needed, rephrase accordingly.

ANSWER – the sentence has been rephrased: The specific properties of water, in particular buoyancy that reduces the effect of weight bearing on skeletal joints, and the great density of water that provides workload along all range of movements, could make this environment suitable for different participants.

Ln49- “.].” – Delete this.

ANSWER - it has been removed.

Ln51 – “it is thus” - Please rephrase this as "Thus, it is important"

ANSWER – the sentence has been rephrased as suggested.

Ln52 – “produce” - Please consider using another term here.

ANSWER - “produce” has been substituted by “achieve”.

Ln66-68 - Please consider rephrasing the entire sentence in order to improve readability.

ANSWER - the period has been rephrased: In typical water fitness classes, participants perform complex and highly variable movement sequences, and the intensity of the exercises can vary quite considerably throughout the lesson [21]. The exercise’s intensity can be controlled by setting both the rhythm and the range of execution during aquatic resistance movements [22].

Ln73 – “results” - Please either rephrase the sentence or use another term here.

ANSWER - the sentence has been rephrased: When walking on a treadmill with subjective speed control, muscle activation of the lower limbs in the water is higher than on land [3].

Ln84-87 - this sentence needs to be rephrased.

ANSWER - the sentence has been rephrased: During speed-controlled water exercises, the direct assessment of heart rate and oxygen uptake levels relative to individual maximal values, could provide useful indications for individualised exercise prescriptions.

Ln120 – “minutes” – minute.

ANSWER – the change has been done.

Ln148 – “at the” – from the.

ANSWER – the change has been done.

Ln149 – “(LactatePro2….) “- This is already mentioned earlier. Please remove it.

ANSWER - it has been removed.

Ln151 – “every” – with each.

ANSWER – the change has been done.

Ln158 – statistical analysis - There is no information of statistical software used for this study. Please add.

ANSWER – the information has been added: All statistical tests were performed using the software SPSS version 20.0 (SPSS, Chicago, IL, USA) and Microsoft Excel 2010.

Ln165 – p-value needs to be lower case letter. p<0.05.

ANSWER – the change has been done.

Ln174 – “statistically” – significantly.

ANSWER – the change has been done.

Ln175 – “already” - please check if this term is appropriate.

ANSWER – “already” has been replaced with “even”.

Ln177 – Mean - capital letter.

ANSWER – the change has been done.

Ln185 – “with respect to” - compared to.

ANSWER – the change has been done.

Ln186 – “guidance” – guidelines.

ANSWER – the change has been done.

Ln188 – “still were” – were still.

ANSWER – the change has been done.

Ln202 – “xyphoid” – xiphoid.

ANSWER – the change has been done.

Ln206 – American College of Sports Medicine – check everywhere in the text and correct.

ANSWER – the change has been done and checked all over the text.

Ln213-215 - This sentence needs to be rephrased.

ANSWER - the sentence has been rephrased: Running suspended requested the least cardiometabolic effort; blood lactate accumulation was above resting values, ranging from 52% to 76% of HRmax, from 49% to 72% of VO2max, with a blood lactate accumulation from 2 to 5 mM/l. Its heart rate values were included in the moderate intensity domain at all exercise frequencies, whilst VO2% achieved the vigorous intensity domain at 60 cpm.

Ln223 – “agree” - Please check if this term is appropriate; Ln224 – “speeds” – speed; Ln225 – “was” - use another term here, please.

ANSWER – the sentence has been rephrased as follows: These results agree with those reported throughout a 40 min water exercise trial, when the greatest motion speed possible was reached whilst maintaining a full range of motion, and the heart rate reached 90% HRmax.

Ln229 - “movement’s” – movement.

ANSWER – the change has been done.

Ln234-237 – Rephrasing the sentence is suggested.

ANSWER – the sentence has been rephrased as follows In water running, the possibility for all the participants to regulate their individual metabolic requirement for each rhythm of exercise could lead to a greater exercise adherence for fitness and health purposes [4].

Ln244 – “running suspended” – Do you mean suspended running?

ANSWER – yes, we meant suspended running and it has been amended in the text.

Ln245,246 - Rephrasing this part of the sentence is necessary.

ANSWER – the sentence has been rephrased: On the other hand, when running in the water at the xiphoid level of immersion, the resistance of the water could hinder the attainment of ample ranges of motion and vigorous intensities of exercise.

Ln273 – “means”  - Please use "tool" or another similar term.

ANSWER – “means” has been replaced with “method”.

Ln286 – “predetermined load…” - Please recheck this part of the sentence, and rephrase it to improve readability.

ANSWER – the sentence has been rephrased: Due to the possibility to control a motion’s rhythm and amplitude, water cycling appears to be a more suitable exercise modality over water running for athletes’ training purposes. It allows them to respect the predetermined load and to practise at vigorous exercise in-tensities.

Ln287 – “coached” – coaches.

ANSWER – the change has been done.

Ln291 – “tests” – test.

ANSWER – the change has been done.

Ln292 – “troubles” - Please use a different term here.

ANSWER – “troubles” has been replaced with “complications”.

Ln296 – “significantly” – This is not necessary, can be removed.

ANSWER - it has been removed.

Ln298 – “bout” – This is not necessary, can be removed.

ANSWER - it has been removed.

Ln301 – “in” – Do you mean at steady state?

ANSWER – “in” has been replaced with “at”.

Ln302-305 – Please rephrase in order to improve readability.

ANSWER – the sentence has been rephrased as follows It could therefore be suggested that water cycling could represent an appropriate exercise mode for training purposes, as it enables athletes to reach vigorous intensities of exercise.

Ln306,307 – “Limits of the present study are, certainly the low number of participants and the whole female representative” - Rephrasing this part is suggested.

ANSWER – the sentence has been rephrased Amongst the limitations of the present study are the restricted number and characterization of the participants, being representative of Caucasian, young and healthy females only. Moreover, exercises have been tested at a single water depth, and at submaximal speeds of motion only.

Ln309 – “brought about” – Please use another more appropriate term here.

ANSWER – “brought about” has been replaced with “obtained”.

Ln311,312 - Please try to write two sentences here instead of one, and rephrase in order to improve the readability.

ANSWER – the sentence has been rephrased as follows: In conclusion, the three water exercise modalities analysed, i.e., cycling, suspended running and running in contact with the bottom of the pool, provide different intensities at the same speed of motion. Water cycling elicited the highest energy consumption, whilst sus-pended water running requested the least cardiometabolic effort and presented the lowest blood lactate accumulation.

Ln316 – “highly fit” - Check if this term is appropriate and rephrase if needed.

ANSWER – “highly fit” has been replaced with “athletes”.

Round 2

Reviewer 1 Report

It is my understanding from the statement "Blood lactate accumulations, calculated subtracting the rest from the peak values obtained during the recovery phase,..." that the blood lactate value presented are the changes that occurred. Please clarify that in Table 1 (and throughout the manuscript) as it now reads for Table 1 that absolute values are presented but they are a change of blood lactate. 

Author Response

Dear Reviewer,

thank you for pointing out the need for clarification, we have modified table 1 header and footer.

Table 1 shows that the heart rate and oxygen consumption of suspended running resulted significantly lower than the hydrobike in each exercise intensity. Suspended running also presented a significantly lower heart rate than running with ground contact at 50 and 60 cpm and a significantly lower oxygen consumption at 40, 50 and 60 cpm. Blood lactate accumulations, calculated subtracting the rest from the peak values obtained during the recovery phase, resulted significantly lower for suspended running compared to the hydrobike at 50 and 60 cpm only. In the HB exercise bLa exceeded 4 mM/l even at 50 cpm, whilst water running requested bLa above 4 mM/l only at 60 cpm.

Table 1. Mean and standard deviation values for each exercise intensity and modality.

30 cpm

40 cpm

50 cpm

60 cpm

HR (bpm)

HB

115 ± 12

140 ± 15

170 ± 19***

188 ± 18***

RC

110 ± 13

134 ± 15

149 ± 18**

166 ± 17**

RS

103 ± 6*

123 ± 11*

138 ± 8*

149 ± 188*

VO2 (ml·kg-1·min-1)

HB

24.8 ± 3.2

28.1 ± 3.5

34.8 ± 3.4***

38.1 ± 3.6***

RC

23.8 ± 2.2

26.5 ± 2.5**

30.3 ± 2.6**

33.7 ± 2.9**

RS

22.0 ± 2.0*

25.3 ± 3.3*

28.7 ± 3.3*

32.6 ± 3.5*

bLa (mM/l)

HB

2.2 ± 0.4

2.8 ± 0.2

4.6 ± 0.9***

6.9 ± 0.7***

RC

2.2 ± 0.3

2.8 ± 0.3

3.6 ± 0.5

6.1 ± 0.6

RS

2.1 ± 0.2

2.6 ± 0.1

3.2 ± 0.4*

5.0 ± 0.9*

*=p<0.05 between RS and HB; **=p<0.05 between RS and RC; ***=p<0.05 between HB and RC. HR: heart rate; VO2: oxygen consumption; bLa: blood lactate accumulation. RS: running suspended; RC: running in contact with the bottom of the pool; HB: hydrobike.
